# Visual Implicit Autoregressive Modeling

Pengfei Jiang [1 2 *]   Jixiang Luo [1 *]   Luxi Lin [2]   Zhaohong Huang [2]   Xuelong Li [1]

## Abstract

Visual Autoregressive Modeling (VAR) based on next-scale prediction achieves strong generation quality, but their explicit deep stacks fix the amount of computation per scale and inflate memory at high resolutions. We introduce Visual Implicit Autoregressive Modeling (VIAR), a next-scale autoregressive generator that embeds an implicit equilibrium layer between shallow pre/post blocks. The implicit layer is trained with Jacobian-Free Backpropagation, yielding constant training memory, while inference exposes a per-scale iteration knob that enables compute control. On ImageNet $256 \times 256$ benchmark, VIAR attains FID 2.16, and sFID 8.07 with only 38.4% parameters of VAR, matching or surpassing strong AR baselines and remaining competitive with large diffusion models. By controlling the per-scale knob, VIAR can reduce peak memory from 19.24 GB to 8.53 GB and doubles throughput from 15.16 to 32.08 images/s on a single RTX 4090, without retraining. Ablations show that fewer steps are sufficient for fixed-point iterations to converge and that VIAR consistently dominates VAR across quality efficiency operating points. In zero shot in-painting and class-conditional editing, VIAR produces sharper details and smoother boundaries while preserving global structure, validating the benefits of implicit equilibria and per-scale compute control for practical, deployable visual generation. Our code and models are available at https://github.com/mobiushy/VIAR.

## 1. Introduction

Modern image generation has been dominated by two architectural paradigms: diffusion models and autoregressive

*Equal contribution [1]Institute of Artificial Intelligence (TeleAI), China Telecom [2]Xiamen University. Correspondence to: Xuelong Li <xuelong_li@ieee.org>.

*Proceedings of the $43^{rd}$ International Conference on Machine Learning*, Seoul, South Korea. PMLR 306, 2026. Copyright 2026 by the author(s).

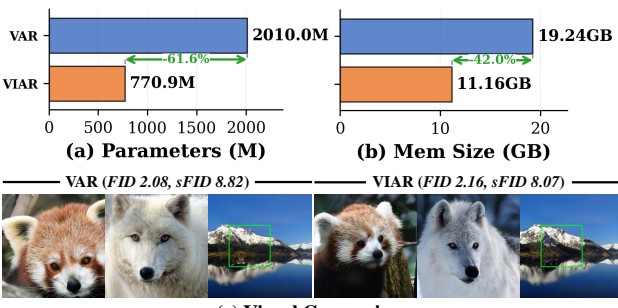

*Figure 1.* Resource savings of VIAR versus VAR. (a) By collapsing the explicit middle stack into a single implicit layer, VIAR reduces the total parameters by 61.6% and the middle-block parameters by 93.3%. (b) Its implicit design with adaptive per-scale iteration further lowers the GPU memory size during parallel next-scale prediction by 42.0%. (c) Despite this extreme lightweight configuration, VIAR maintains strong generation and editing quality.

(AR) models. While diffusion models (Ho et al., 2020; Song et al., 2020; Peebles & Xie, 2023; Rombach et al., 2022) have set a high bar for image quality, recent AR (Esser et al., 2021; Yu et al., 2021; Razavi et al., 2019; Lee et al., 2022; Yu et al., 2022) advances have substantially narrowed the gap by leveraging transformers at scale and stronger visual tokenizers. However, conventional next-token AR on images inherits a fundamental mismatch between 1D causal ordering and 2D spatial structure: flattening destroys locality, induces bidirectional correlations in practice, and leads to inefficient sampling (Van den Oord et al., 2016). Visual Autoregressive modeling (VAR) (Tian et al., 2024) addresses these issues by redefining autoregression as next-scale prediction: a model generates multi-scale token maps from coarse to fine, with full parallelism within each scale. This hierarchical, coarse-to-fine ordering preserves spatial locality, reduces computational complexity relative to raster-scan AR, and exhibits promising scalability and zero-shot generalization (e.g., in-/out-painting and editing). Yet VAR retains a deeply stacked explicit backbone within each next-scale step. As model width and image resolution increase, memory footprint, latency, sampling depth, and parallelism come into tension. Moreover, fixed-depth computation per scale limits the compute-on-demand behavior required for mobile or edge deployment and low-latency online serving.

A key observation from deep equilibrium models (DEQs) is directly relevant here. DEQs (Bai et al., 2019; 2020;

Geng & Kolter, 2023) replace deep explicit stacks with an implicit fixed-point layer and train via implicit differentiation or Jacobian-free backpropagation (Geng et al., 2021; Fung et al., 2022), enabling constant-memory backpropagation, controllable convergence, and variable compute at test time. Embedding such implicit layers in iterative generative processes unlocks new levers, enabling the reallocation of computation across steps and the reuse of solutions across closely related subproblems to achieve superior balance between quality and cost under fixed budgets (Bai & Melas-Kyriazi, 2024; Pokle et al., 2022; Geng et al., 2023). These properties suggest that integrating deep equilibrium implicit layer into VAR can preserve VAR's strengths while overcoming the high parallel computational complexity inherent in its fixed-depth next-scale prediction paradigm.

We introduce Visual Implicit Autoregressive Modeling, the first framework to integrate DEQ into VAR's next-scale prediction paradigm. Concretely, at each next-scale step we replace the deep explicit middle stack with a single implicit fixed-point layer, yielding an implicit visual autoregressive architecture. The implicit layer models the effect of an infinitely deep, weight-tied transform conditioned on the input injection at that scale. Training uses Jacobian-free equilibria with constant-memory backpropagation, and inference adaptively adjusts iteration counts per scale. As shown in Figure 1, this design brings two immediate benefits. First, it reduces memory and parameter overhead by collapsing the deep middle into a single implicit operator, while preserving VAR's coarse-to-fine factorization and within-scale parallelism. Second, it decouples compute from depth: per-scale computation becomes a knob (number of fixed-point iterations) rather than a fixed stack, enabling flexible control of compute across scales.

Previous researches (Chen et al., 2025b; Guo et al., 2025) has shown that VAR suffers from significant computational redundancy in large scale predictions. Explicitly stacking intermediate layers leads to excessive model parameters and generated KV cache during parallel token map computation. Our method, however, drastically compresses intermediate layer parameters, and thanks to the ability to customize the number of iterations for each scale, it significantly reduces the generated KV cache. Furthermore, by transitioning from a single forward pass through multiple explicit layers to multiple forward iterations of a single implicit layer, the model depth becomes a flexible iteration count adjustable at inference time rather than a fixed hyperparameter determined before training. This shift means that a single trained VIAR model can effectively emulate networks of varying depths, offering the elastic inference capabilities that previously required training multiple distinct models.

Taken together, VIAR preserves the virtues of next-scale autoregression, such as spatial locality, parallelism within

scales, and favorable complexity, while importing the central strengths of implicit equilibrium layers including constant memory, adaptive compute, and convergence control. Consequently, this synthesis provides a practical path toward scalable, deployable visual generation with strong quality efficiency trade-offs. VIAR is especially advantageous in resource-constrained or low-latency settings, where adaptive per-scale iteration counts enable compute-on-demand deployment to edge devices.

Our contributions are as follows:

• We propose VIAR, a deep equilibrium implicit generator with multi-scale factorization. VIAR delivers constant training memory and pre-scale variable compute, enabling efficient training and flexible inference.

• We leverage VIAR to implement flexible compute control cross scales, reducing redundant computation at larger scales. This yields substantial memory savings and better quality latency trade-offs under restricted compute budget.

• Empirically, VIAR matches the FID, sFID, and IS of strong VAR baselines while using markedly fewer total parameters (approximately 38.4% of the original), and it retains strong zero-shot inpainting and editing capabilities.

## 2. Related Work

### 2.1. Efficient Visual Autoregressive Generation

Work on making visual autoregressive generation efficient follows three complementary threads. First, decoding parallelization replaces strictly serial steps with collaborative or speculative procedures: collaborative decoding amortizes context exchange to cut memory and yield about 1.7× speedups with negligible FID change (Chen et al., 2025b). Speculative/Jacobi-style methods parallelize candidate expansions with consistency checks, including training-free SJD, grouped speculative decoding, and maximal-coupling or denoising-aided variants for stronger stability and quality preservation (Teng et al., 2024; So et al., 2025b;a; Teng et al., 2025), but these methods are specifically designed for autoregressive image generation for the next token prediction paradigm. Second, large-scale redundancy is reduced by computing only where it matters: cached-token pruning makes resolution scaling near-linear by forwarding pivotal tokens and restoring pruned slots from caches (Guo et al., 2025), stage-aware allocation shifts budget across scales (Li et al., 2025c), frequency-aware sparsification drops low-frequency tokens at high resolutions (Chen et al., 2025a), and adaptive skipping accelerates inference without retraining (Li et al., 2025a). Third, system-level memory pressure is addressed by scale-aware KV cache compression and reconstruction, which shrinks peak cache while retaining cross-scale context fidelity (Li et al., 2025b).

## 2.2. Deep Equilibrium Models

DEQs replace explicit deep stacks with an implicit fixed-point layer trained via implicit or Jacobian-free differentiation, enabling constant-memory backpropagation and adaptive compute (Bai et al., 2019). Subsequent work extends this framework with hierarchical operators and scale-coupled equilibria, improving both optimization stability and representational capacity (Bai et al., 2020). Jacobian-Free Backpropagation (JFB) provides a practical 1-step gradient through a single unrolled iteration, with subsequent multi-step and stochastic variants improving stability–bias trade-offs in large-scale settings (Fung et al., 2022). In diffusion, equilibrium formulations have been explored at two complementary granularities. On the one hand, by solving for entire diffusion trajectories or constructing equilibrium surrogates for the reverse process (Pokle et al., 2022), on the other hand, by inserting an implicit fixed-point layer into the denoiser so that each timestep is solved to equilibrium, enabling timestep-wise compute reallocation and solution reuse under fixed budgets (Bai & Melas-Kyriazi, 2024). Beyond diffusion, DEQ layers have been applied to dense prediction and correspondence problems (e.g., optical flow) demonstrating that equilibrium inference can replace deep iterative refinements while preserving accuracy and reducing memory (Bai et al., 2022). Taken together, these works highlight three themes that we exploit in a next-scale AR setting. First, implicit layers decouple compute from depth through convergence-controlled iterations. Second, JFB-style gradients make constant-memory training practical. Third, in iterative generators, reusing equilibria and reallocating iterations across steps delivers better balance between quality and efficiency than fixed-depth stacks.

## 3. Method

### 3.1. Preliminaries

**Next-Scale Autoregressive Modeling.** VAR reframes image autoregression from next-token to next-scale prediction over a hierarchy of discrete token maps in two stages. This shift preserves spatial structure and enables parallel prediction, setting up a coarse-to-fine generation process.

In the multi-scale tokenization stage, an image $im$ is encoded to a latent feature map $f = E(im)$ and quantized across $K$ resolutions with a shared codebook $Z$:

$$r_k(i,j) = \arg \min_{v \in [V]} \| Z_v - \text{interp}_k(f)(i,j) \|_2^2, \quad (1)$$

where $k = 1, \ldots, K$. Decoding looks up codes and reconstructs with VAE decoder $D$. In practice, the tokenizer is trained with a compound VQ objective to balance fidelity and perceptual quality.

Given the token hierarchy, VAR models the joint distribution

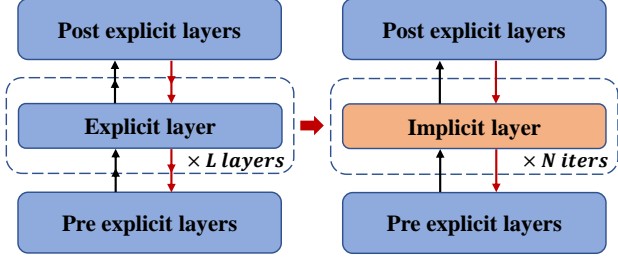

**(a) Comparison between VIAR and VAR**

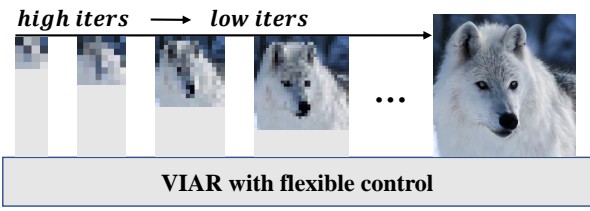

**(b) VIAR Generation Process**

*Figure 2.* Overview of VIAR. (a) The Implicit Architecture. We replace the deep explicit VAR stack with a single implicit equilibrium layer between shallow pre/post-layers. Training via Jacobian-Free Backpropagation unrolls only the last $m$ iterations for gradients, ensuring constant memory usage. (b) VIAR Generation Process. During inference, unlike standard VAR which is constrained to fixed computation, VIAR flexibly control compute across scales through an iterations knob, thereby reducing high-resolution redundancy, latency, and KV cache usage while preserving quality.

via a coarse-to-fine factorization, predicting all tokens at a scale in parallel conditioned on previous scales:

$$p(r_1, \ldots, r_K) = \prod_{k=1}^{K} p(r_k \mid r_{<k}), \quad (2)$$

with per-scale prediction instantiated as $\hat{r}_k \sim p_\theta(\cdot \mid r_{<k})$. During training, a block-wise causal mask ensures tokens at scale $k$ attend only to $r_{\leq k}$. At inference time, KV caching preserves this autoregressive dependency without the overhead of explicit masking. Within each scale, self-attention incorporates 2D positional signals and applies the block causal mask $M$ to enforce causality:

$$\text{Attn}(Q, K, V; M) = \text{softmax}\big((QK^\top + M)/\sqrt{d}\big)V, \quad (3)$$

This design maintains spatial coherence, supports within-scale parallelism, and provides a clean interface for adaptive compute policies introduced later.

In terms of complexity, under the native AR setting without KV caching, raster-scan next-token AR over $n \times n$ tokens requires $O(n^6)$ total self-attention operations. Next-scale prediction reduces this to $O(n^4)$ by enabling within-scale parallelism and limiting the number of autoregressive steps. For example, in next-scale prediction, if the resolutions follow $n_k = a^{k-1}$ with $a > 1$ and $n_K = n$, the total compute over all scales becomes $\sum_k n_k^4 = O(n^4)$.

## 3.2. Deep Equilibrium Visual Autoregressive Modeling

The overall framework of VIAR is shown in Figure 2. Consider the $k$-th scale with conditioning on all previous scales $r < k$ and optional class or task condition $c$. Let $H_k$ denote the hidden feature space at scale $k$. We apply $p$ explicit pre-blocks to project and normalize the inputs into an injected representation, solve a fixed-point problem in an implicit layer, and then apply $p$ explicit post-blocks for residual transformation and prediction.

**Pre-projection and input injection**. We define a pre-transform $f_{\text{pre}}(\cdot; \theta_{\text{pre}})$, implemented as a composition of $p$ transformer blocks with attention and FFN modules. Given the last scale output $e_{k-1}$ as input, we compute

$$x_k = f_{\text{pre}}(e_{k-1}, c; \theta_{\text{pre}}) \in H_k, \qquad (4)$$

which serves as an input injection to the implicit layer.

**Implicit equilibrium layer**. We define a contractive map $f_{\text{imp}}(\cdot; \theta_{\text{imp}})$ that, given the injected input $x_k$ and a hidden state $z_k \in H_k$, produces an updated hidden state. Starting from an initialization derived from $x_k$, the equilibrium $z_k^*$ is the fixed point of this map, i.e., the solution satisfying $z_k^* = f_{\text{imp}}(z_k^*; \theta_{\text{imp}})$ under the conditioning on $x_k$:

$$z_k^* = f_{\text{imp}}(z_k^*, x_k, c; \theta_{\text{imp}}), \qquad (5)$$

which we compute by fixed-point iteration or quasi-Newton methods. In practice, we concatenate $z_k^*$ and $x_k$ and use a projection to fuse representations before entering the implicit transformer block:

$$\tilde{z} = \text{Proj}([z_k^*, x_k]), \qquad z_k^* = f_{\text{blk}}(\tilde{z}, c; \theta_{\text{blk}}). \quad (6)$$

Where $f_{\text{blk}}$ is a transformation defined by a single transformer block.

**Post-projection**. After equilibrium is reached, we pass $z_k^*$ to a post-transform $f_{\text{post}}(\cdot; \theta_{post})$, implemented as another composition of $p$ transformer blocks, to post projection the equilibrium state:

$$\hat{r}_k = f_{\text{post}}(z_k^*, c; \theta_{\text{post}}). \qquad (7)$$

The pre-projection and post-projection provide stable interfaces for conditioning and geometric/semantic alignment across scales, while the implicit layer captures infinite depth refinement controlled by the convergence tolerance or iteration budget. Thus, we trade explicit depth for iteration-controlled compute. This yields an adaptive per-scale compute knob via the number of fixed-point iterations.

### 3.3. Stochastic Jacobian-free Backpropagation

We maximize the next-scale likelihood using standard cross-entropy under the factorization $p(r_1, \ldots, r_K) = \prod_k p(r_k \mid$

---

**Algorithm 1** VIAR Training with S-JFB

1: **Input:** token embedding $e$, conditioning $c$, parameters $\Theta = \{\Theta_{\text{pre}}, \Theta_{\text{imp}}, \Theta_{\text{post}}\}$, bounds $N, M$
2: **Pre-layers:** $x \leftarrow f_{\text{pre}}(e, c; \Theta_{\text{pre}})$
3: **Input injection:** $x_{\text{inj}} \leftarrow \text{clone}(x)$
4: **Initialize:** $z \leftarrow x$ {warm start}
5: **Sample no-grad iters:** $n \sim \text{Uniform}\{0, \ldots, N\}$
6: **for** $t = 1$ **to** $n$ **do**
7:     {no gradient accumulation}
8:     $z \leftarrow G(z; x_{\text{inj}}, c) \triangleq f_{\text{imp}}(\text{Proj}([z, x_{\text{inj}}]), c; \Theta_{\text{imp}})$
9: **end for**
10: **Sample with-grad iters:** $m \sim \text{Uniform}\{1, \ldots, M\}$
11: **for** $t = 1$ **to** $m$ **do**
12:     {record graph for backprop}
13:     $z \leftarrow G(z; x_{\text{inj}}, c)$
14: **end for**
15: **Post-layers:** $\hat{r} \leftarrow f_{\text{post}}(z, c; \Theta_{\text{post}})$
16: **Compute loss:** $\mathcal{L} \leftarrow \text{CrossEntropy}(\hat{r}, r)$
17: **Parameter updates:**
18:     Update $\Theta_{\text{pre}}, \Theta_{\text{post}}$ by standard backpropagation
19:     Update $\Theta_{\text{imp}}$ with the S-JFB gradient by backpropagating only through the last $m$ implicit iterations

---

$r_{<k}$). Let $\mathcal{L}_k$ denote the token-level negative log-likelihood at scale $k$, and let the total loss be defined as:

$$\mathcal{L} = \sum_{k=1}^{K} \mathcal{L}_k = -\sum_{k=1}^{K} \log p_\theta(\hat{r}_k \mid r_{<k}). \qquad (8)$$

Direct implicit differentiation through $z_k^*$ can be accurate but memory-expensive. We adopt stochastic Jacobian-free backpropagation (S-JFB) to obtain stable gradients with near-constant memory. For each training step and each scale $k$, we sample $n \sim U\{0, \ldots, N\}$ and $m \sim U\{1, \ldots, M\}$, perform $n$ fixed-point iterations without gradient tracking to approach equilibrium, then perform $m$ additional iterations with gradient tracking, and backpropagate only through the last $m$ iterations. Formally, if $G(z; x_k, c) := f_{\text{imp}}(z, x_k, c; \theta_{\text{imp}})$, the iteration is

$$z_k^{t+1} = G_{\text{ng}}(z_k^t; x_k, c), \quad t = 0, \ldots, n-1, \qquad (9)$$

and

$$z_k^{t+1} = G_{\text{wg}}(z_k^t; x_k, c), \quad t = n, \ldots, n+m-1. \quad (10)$$

Let $\hat{z}_k$ denote the final iterate after $n + m$ steps. We approximate the true gradient of $\theta_{\text{imp}}$ by backpropagating through the last $m$ unrolled steps only:

$$\frac{\partial \mathcal{L}}{\partial \theta_{\text{imp}}} \approx \sum_{k=1}^{K} \frac{\partial \mathcal{L}_k}{\partial \hat{z}_k} \cdot \frac{\partial \hat{z}_k}{\partial \theta_{\text{imp}}}\Big|_{\text{last } m \text{ steps}}. \qquad (11)$$

Algorithm 1 demonstrates the specific teacher-forcing training process of VIAR. This stochastic multi-step JFB strikes

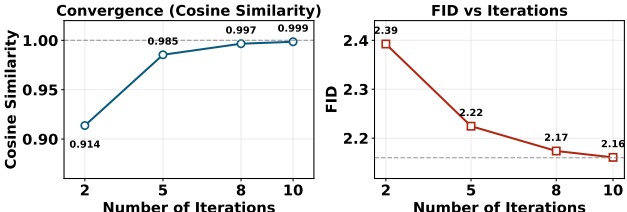

*Figure 3.* Convergence analysis of the implicit equilibrium layer at high-resolution scales. We measure the cosine similarity between the last two steps of the iteration at the largest scale. The similarity achieves 0.985 by iteration 5 and approaches 0.999 by iteration 10, indicating that the fixed-point iteration converges rapidly. The corresponding FID changes on the right further support this point. This rapid convergence justifies reducing iteration counts at large scales to save compute without significant quality loss.

a practical balance between stability and efficiency. Using uniform sampling $m$ reduces the bias inherent to purely 1-step gradients while lowers memory footprint. By contrast, excessively large $m$ (or $n$) tends to slow training without clear benefits and can even destabilize early optimization when the operator's local Lipschitz constant is large.

### 3.4. Multi-scale Sampling with Compute Control

Next-scale generation proceeds from coarse to fine. Empirically, large (high-resolution) scales often focus on refining details, where full-depth explicit stacks are overkill. Moreover, parallel within-scale prediction dramatically increases memory cost at high resolutions. Our implicit equilibrium layer exposes a natural per-scale compute knob, the iteration count, which can be reduced at large scales or reallocated to earlier scales to improve global structure.

Let $K$ be the total scales and let $c_k$ be the number of fixed-point iterations at scale $k$. The total compute budget of the implicit autoregressive modeling is

$$\mathcal{C} = \sum_{k=1}^{K} \Big( p_{\text{pre}} + c_k + p_{\text{post}} \Big), \qquad (12)$$

where $p_{\text{pre}} = p_{\text{post}} = p$. Under a fixed budget $\mathcal{C}$, we can choose $\{c_k\}$ to minimize validation loss or a proxy, subject to nonnegativity and practical bounds:

$$\min_{\{c_k\}} \ \sum_{k=1}^{K} \mathbb{E}\big[\mathcal{L}_k(c_k)\big] \quad \text{s.t.} \quad \sum_{k=1}^{K}(p + c_k + p) \leq \mathcal{C}. \quad (13)$$

We control per-scale compute using heuristics or adaptive schedules. The constant schedule sets $c_k = c$ for all $k$. The decreasing schedule assigns more iterations to coarser scales to stabilize global structure and fewer to finer scales to save compute. Another simple strategy is adaptive threshold control, which stops at scale $k$ when $\|G(z; x_k, c) - z\|_2 \leq \tau_k$, with $\tau_k$ selected to satisfy the global budget on average.

Since the number of iterations (rather than depth) is the unit of cost, these schedules can directly reduce computationally intensive high-resolution steps.

Even with fewer iteration steps at high-resolution scales, VIAR can still achieve competitive generation quality due to the rapid convergence of the implicit solver. The convergence analysis in Figure 3 shows that the implicit layer at the largest scales reaches cosine similarity greater than 0.98 by 5 iterations and almost converges after 10 iterations, indicating rapidly saturating refinements. Therefore, the iteration budgets at high-resolution scales can be substantially reduced. This iteration-level control eliminates redundant computation at large scales and achieves a better quality–cost balance with negligible loss of fidelity.

With next-scale factorization and within-scale parallel prediction, replacing raster-scan next-token AR reduces total attention compute from $O(n^6)$ to $O(n^4)$. Our deep equilibrium architecture does not alter this asymptotic gain and further reduces training and inference cost by introducing an implicit layer between shallow pre and post blocks, effectively decoupling memory from model depth.

## 4. Experiment

### 4.1. Experimental Setup

We implement VIAR on a 2B-parameter VAR with the original multi-scale VQVAE tokenizer (Tian et al., 2024; Razavi et al., 2019). The tokenizer is frozen during AR training. Each scale uses three modules: pre-layers, an implicit layer, and post-layers. All share the same transformer block design except for the implicit layer adds one projection for input injection. We set $p = 5$ blocks for both pre-layers and post-layers. We adopt S-JFB with $N = 10$ no-grad and $M = 12$ with-grad iterations by default (Fung et al., 2022). Global batch size and learning rate are 512 and 8e-5, respectively. Other optimization settings follow the VAR configuration. More implementation details can be found in the appendix. For evaluation, we evaluate class-conditional ImageNet at 256×256 resolution. Metrics are computed on 50k generated samples against the ImageNet training set following ADM (Dhariwal & Nichol, 2021) and include Inception Score (IS) (Salimans et al., 2016), Fréchet Inception Distance (FID) (Heusel et al., 2017), spatial Fréchet Inception Distance (sFID), Precision (Pre), and Recall (Rec). We also report parameter count, peak GPU memory, and throughput to characterize efficiency.

### 4.2. Main Results

Table 1 presents the quantitative comparison between VAR and VIAR on ImageNet 256×256. At a classifier-free guidance (CFG) scale of 2.0, VIAR achieves a superior Inception Score of 330.7 compared to VAR-d30, and significantly im-

*Table 1.* Quantitative comparison between VAR and VIAR on ImageNet $256 \times 256$. We report Inception Score (IS), Fréchet Inception Distance (FID), spatial Fréchet Inception Distance (sFID), Precision (Pre), Recall (Rec), parameters (#Params), and Inference memory overhead (#Infer mem). **VIAR** achieves competitive performance with significantly fewer parameters compared to the baseline VAR.

| MODELS | FID ↓ | sFID ↓ | IS ↑ | PRE ↑ | REC ↑ | #PARAMS | #INFER MEM |
|---|---|---|---|---|---|---|---|
| VAR-D30 (CFG=2.0) | **2.05** | 8.86 | 328.5 | 0.82 | 0.59 | 2010.0M | 19.24GB |
| VAR-D30 (CFG=1.5) | 2.08 | 8.82 | 306.8 | 0.82 | 0.59 | 2010.0M | 19.24GB |
| **VIAR (CFG=2.0)** | 2.35 | **7.92** | **330.7** | **0.83** | 0.58 | 770.9M | 11.16GB |
| **VIAR (CFG=1.5)** | 2.16 | 8.07 | 300.1 | 0.81 | **0.59** | 770.9M | 11.16GB |

*Table 2.* Adaptive per-scale iteration schedules on ImageNet-256. Each row sets the iteration counts for coarse and fine scales. Results show two consistent trends: (1) fixed-point iteration at each scales effectively converges around 10 steps. (2) when the large scales are not yet converged, increasing iterations at the small scales improves FID. Decreasing schedules that shift compute to coarse scales therefore provide a better quality efficiency balance.

| SCHEDULE | FID ↓ | sFID ↓ | IS ↑ | PRE ↑ | REC ↑ |
|---|---|---|---|---|---|
| $Dec._{(20,5)}$ | 2.18 | 8.04 | 299.2 | 0.81 | 0.58 |
| $Dec._{(20,10)}$ | 2.16 | 8.07 | 294.8 | 0.81 | 0.59 |
| $Dec._{(10,5)}$ | 2.22 | 8.08 | 303.4 | 0.82 | 0.58 |
| $Con._{(20,20)}$ | 2.16 | 8.17 | 294.8 | 0.81 | 0.59 |
| $Con._{(5,5)}$ | 2.27 | 8.02 | 307.1 | 0.82 | 0.58 |
| $Con._{(10,10)}$ | 2.16 | 8.07 | 300.1 | 0.81 | 0.59 |

*Table 3.* Throughput and memory comparison on a single RTX 4090. $VIAR_{si}$ denotes VIAR configured with certain iteration schedule, a larger subscript means fewer implicit iterations. As the schedule becomes more aggressive, peak GPU memory and latency improve markedly while image quality remains competitive.

| METHOD | FID ↓ | sFID ↓ | MEM (GB) ↓ | IMAGES/S ↑ |
|---|---|---|---|---|
| VAR | 2.08 | 8.82 | 19.24 | 15.16 |
| $VIAR_{s1}$ | 2.16 | 8.07 | 11.16 | 21.50 |
| $VIAR_{s2}$ | 2.22 | 8.08 | 9.60 | 26.92 |
| $VIAR_{s3}$ | 2.27 | 8.02 | 9.40 | 28.12 |
| $VIAR_{s4}$ | 2.43 | 8.28 | 8.53 | 32.08 |

proves spatial fidelity with an sFID of 7.92 versus VAR's 8.86. Notably, at a lower CFG of 1.5, the FID of VIAR is extremely close to VAR, while maintaining a clear advantage in sFID and achieving its best Recall. These results are delivered with drastically fewer parameters: 770.9M compared to VAR's 2010.0M, a reduction of 61.6%. The ablation of the VIAR implicit architecture and the quantitative comparison results with other AR models and diffusion models are included in the appendix.

Figure 4 provides a qualitative comparison. Samples generated by VIAR exhibit sharpness and semantic coherence on par with or better than VAR, with cleaner textures and fewer artifacts. This visual evidence corroborates the quantitative metrics, confirming that VIAR's implicit equilibrium design efficiently matches the generation quality of deep explicit stacks while significantly reducing model size.

### 4.3. Adaptive Per-scale Iteration Schedules

We investigate how the allocation of implicit iterations at different scales affects the generation quality of VIAR. Let $Dec._{(a,b)}$ is the linear decreasing schedule ($a$ at the smallest scale, $b$ at the largest scale), and $Con._{(a,a)}$ uses the same count at all scales. This directly controls where VIAR invests compute: coarse scales shape global structure, whereas fine scales mainly refine details.

Table 2 and Figure 3 shows that the implicit equilibrium layer stabilizes quickly at each scale. Hence, at inference

time only a small number of fixed-point iterations are needed for accurate refinement. Since the implicit layer's cost scales with the number of iterations, capping the iteration count at large scales substantially reduces latency and memory without sacrificing quality. Another finding is that if the fine-grained stage has not yet converged, increasing the iterations of the coarse-grained stage continuously improve FID, indicating that the details refinement can benefit from additional global structure refinement in the early stage.

Therefore, under tight budgets, the best quality–efficiency balance is achieved by dialing down the iteration count at large scales and reallocating the saved compute to earlier scales. In the experimental section, we adopt $Con._{(10,10)}$ as the default schedule, as it consistently balances quality and efficiency by prioritizing structural convergence early and reducing unnecessary high-resolution iterations.

### 4.4. User-facing Latency and GPU Memory

We benchmark user-facing efficiency on a single RTX 4090, reporting peak GPU memory and end-to-end throughput (images/s) together with FID/sFID to verify quality in Table 3. VIAR adopts four different per-scale iteration schedules, and we vary its aggressiveness from $s1$ to $s4$, where a larger subscript means fewer implicit iterations at all scales.

Across all settings, VIAR consistently reduces memory and increases throughput relative to VAR. With the most aggressive schedule, $VIAR_{s4}$ reaches 32.08 images/s at 8.53 GB, compared to VAR's 15.16 images/s at 19.24 GB. This corresponds to a 2.1× speedup and a 2.26× reduc-

VAR ——————————————— VIAR ———————————————

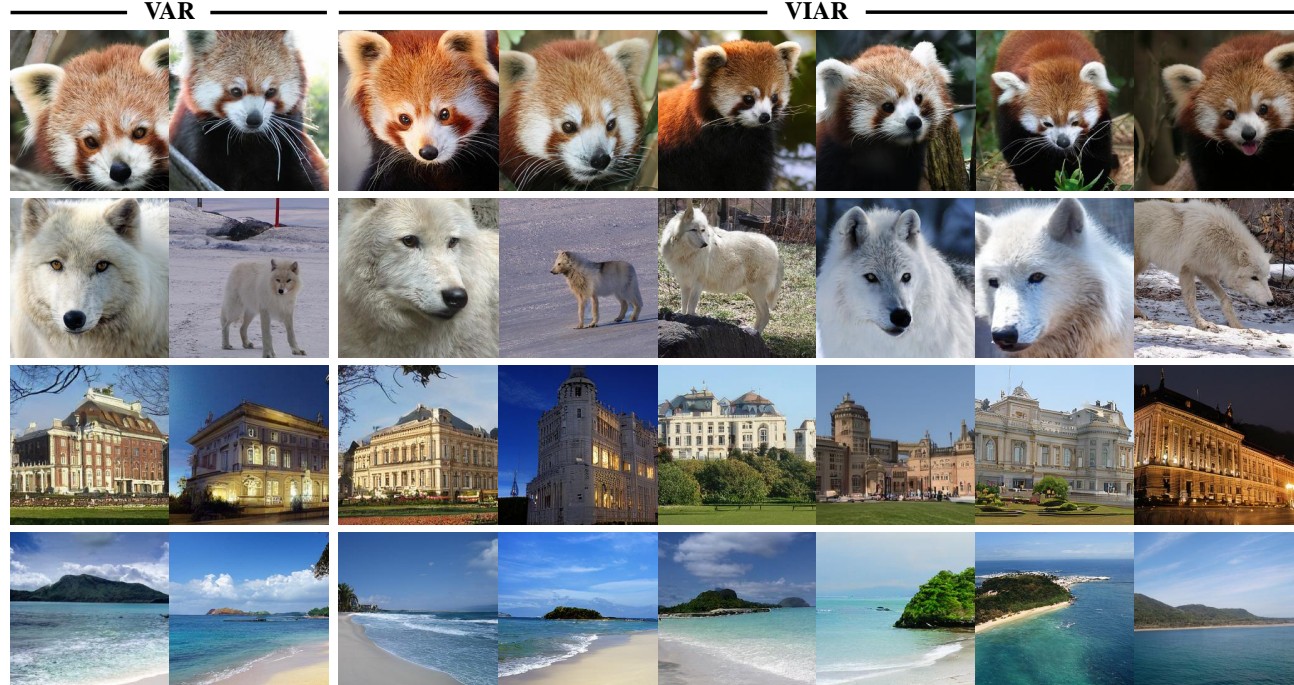

*Figure 4.* Qualitative comparison on ImageNet 256 × 256. For each class we sample with identical decoding settings. Left: VAR; right: VIAR. VIAR matches or surpasses the visual quality of VAR while using fewer parameters and achieving faster inference.

VAR (GPU Mem: 19.24 GB; Throughput: 15.16 images/s)

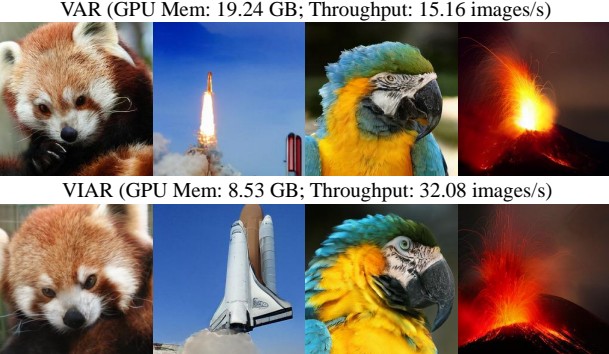

VIAR (GPU Mem: 8.53 GB; Throughput: 32.08 images/s)

*Figure 5.* Visual comparison on ImageNet-256 between VAR and VIAR$_{s4}$. Despite using a more aggressive decreasing schedule, VIAR$_{s4}$ preserves global semantics and textures while running with lower GPU memory and higher throughput.

tion in peak memory, while still maintain competitive quality as shown in Figure 5. Intermediate schedules show a smooth trade-off: VIAR$_{s1}$ already cuts memory to 11.16 GB and raises throughput to 21.50 images/s with the best FID, whereas VIAR$_{s3}$ maximizes sFID with 9.40 GB and 28.12 images/s. As the schedule becomes more aggressive, throughput scales nearly linearly and memory continues to fall, with only minor changes in visual quality.

These gains stem from VIAR's implicit equilibrium layer and flexible compute control: fewer iterations are spent at large scales where refinement is redundant, reducing KV cache growth and activation traffic without retraining. In

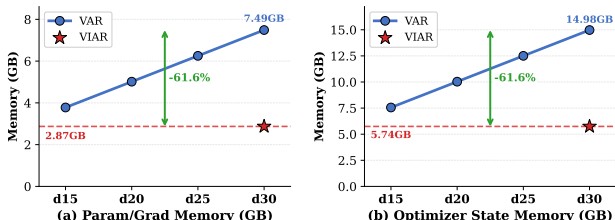

*Figure 6.* Training memory vs. explicit depth. Left: parameter/gradient memory, right: optimizer-state memory. VAR's memory grows roughly linearly with depth, whereas VIAR stays essentially constant because the implicit equilibrium architecture.

practice, $s4$ offers the best latency–memory profile for interactive use, while $s1$ and $s2$ provide the strongest quality under tighter but still favorable budgets. Overall, VIAR delivers a tunable quality–efficiency curve that is better than VAR on a single RTX 4090 deployment.

### 4.5. Constant Training Memory

We next quantify training memory and verify that VIAR's implicit formulation yields depth-independent usage. Figure 6 plots parameter/gradient memory and optimizer-state memory as we increase the explicit depth $d$ of the baseline VAR. VAR's footprint grows roughly linearly with depth because activations and optimizer states scale with the number of stacked blocks. In contrast, VIAR remains largely flat during training due to its implicit design.

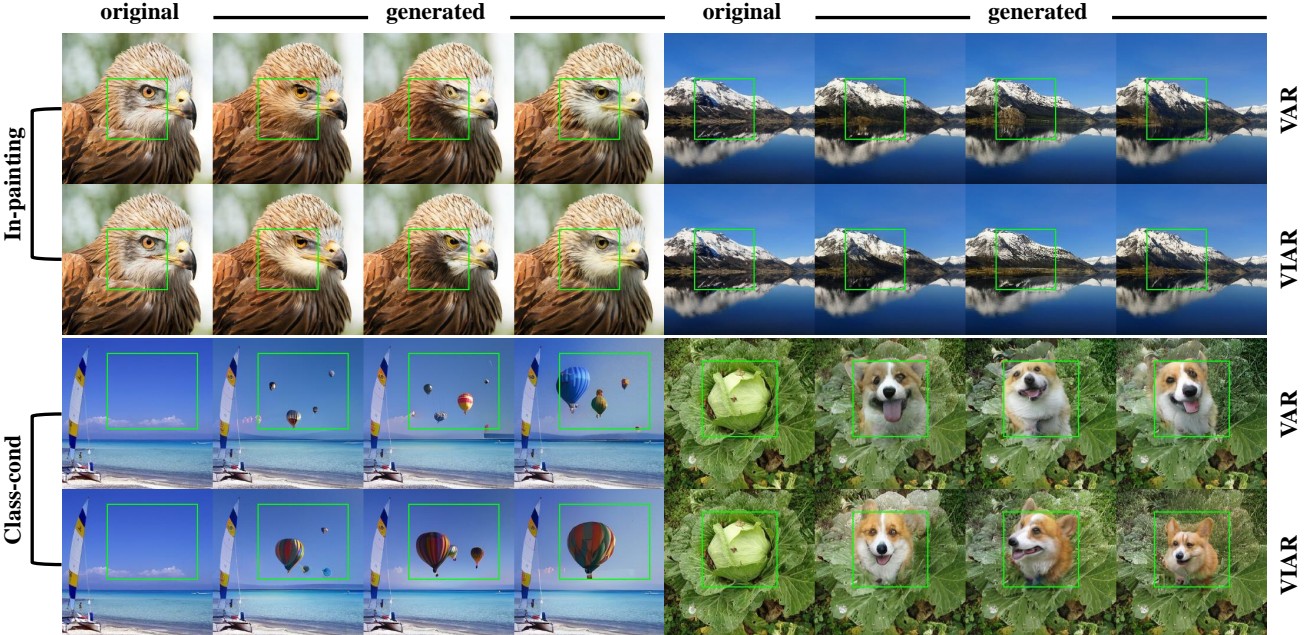

*Figure 7.* Zero-shot generalization of VIAR: in-painting and class-conditional editing without finetuning. For in-painting, tokens outside the mask are teacher-forced and the model generates only the masked region; for class-conditional editing, the model synthesizes content inside a bounding box given a class label while keeping the rest unchanged. Columns show original vs. generated results for VAR and VIAR. VIAR preserves global structure and semantics while producing sharper details and smoother boundary blending, demonstrating stronger zero-shot editing quality with fewer model parameters.

Concretely, VIAR holds at about 2.87 GB (params/grads) and about 5.74 GB (optimizer), while VAR-d30 reaches about 7.49 GB and 14.98 GB, respectively, representing roughly a 61.6% reduction in both views. This constant-memory behavior complements the efficiency gains: by decoupling memory from explicit depth, VIAR allows larger batch sizes or longer sequences on the same hardware, and simultaneously reduces peak training requirements on commodity GPUs without sacrificing quality.

### 4.6. Zero-shot Generalization Editing Tasks

The qualitative comparison results of the zero-shot editing task are shown in Figure 7. In the in-painting setting, tokens outside the mask are teacher-forced and the model generates only the masked region. In the class-conditional setting, the model synthesizes tokens within a user-specified box given a class label while keeping the rest fixed. To ensure a fair comparison of behavior rather than hyperparameters, we keep the tokenizer and decoding settings aligned with the VAR baseline, and then replace VAR with VIAR configured with the default per-scale iteration schedule.

Under this schedule, VIAR achieves rapid convergence at both coarse and fine scales, which reduces the total number of implicit iterations relative to VAR. Despite the lower compute, the edited results are consistently sharper and better integrated with the unedited context. Across diverse masks and categories, VIAR preserves global structure and seman-

tic alignment while producing cleaner textures and more coherent local geometry inside the edited region. The boundaries between edited and original content blend more naturally, and small structures such as feathers, fur, and edges are rendered with fewer artifacts than those produced by VAR. These improvements follow directly from the VIAR design. The implicit equilibrium layer aggregates long-range context, which stabilizes the solution under strong conditioning from the teacher-forced surroundings. Consequently, the system maintains visual quality even when operating under a smaller compute budget.

## 5. Conclusion

We presented VIAR, a visual autoregressive framework that replaces VAR's deep explicit middle stack with a single implicit equilibrium layer trained via Jacobian-Free Backpropagation, and that leverages per-scale iteration schedules to control compute across scales. This design delivers two practical advantages: constant-memory backpropagation during training and flexible, budget-aware inference. Empirically, VIAR achieves competitive ImageNet $256 \times 256$ quality with substantially fewer parameters than large baselines and offers strong user-facing efficiency: reducing the total parameters by 61.6% and peak GPU memory by 42.0% while achieving FID 2.16 and sFID 8.07. VIAR also enhances zero-shot in-painting and class-conditional editing, producing sharper details and cleaner boundary blending.

## Acknowledgements

We would like to thank TeleAI for providing the resources that made this work possible. We are also grateful to all coauthors for their valuable discussions, technical support, and constructive feedback throughout the development of this project. Their contributions greatly improved the design, experiments, and presentation of this paper.

## Impact Statement

This paper presents work whose goal is to advance the field of Machine Learning. There are many potential societal consequences of our work, none which we feel must be specifically highlighted here.

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

---

**Algorithm 2** VIAR Inference (per scale k)

---

1: **Input:** token embedding $e_{k-1}$, conditioning $c$, iteration budget $K_{\text{iter}}$
2: **Pre-layers:** $x_k \leftarrow f_{\text{pre}}(e_{k-1}, c; \Theta_{\text{pre}})$
3: **Input injection:** $x_{\text{inj}} \leftarrow \text{clone}(x_k)$
4: **Initialize:** $z \leftarrow x_k$
5: **for** $t = 1$ **to** $K_{\text{iter}}$ **do**
6:     {no gradient; fixed budget}
7:     $z \leftarrow f_{\text{imp}}(\text{Proj}([z, x_{\text{inj}}]), c; \Theta_{\text{imp}})$
8: **end for**
9: **Post-layers:** $r_k \leftarrow f_{\text{post}}(z, c; \Theta_{\text{post}})$
10: **Return** $r_k$ (and proceed to scale $k+1$)

---

## A. VIAR Inference

Algorithm 2 details the inference procedure for a single scale $k$. The process begins with the token map $e_{k-1}$ from the previous scale (or an initial start token for $k = 1$) and a conditioning signal $c$ (e.g., class embedding). First, the explicit pre-layers $f_{\text{pre}}$ process the input to produce an initial hidden state $x_k$, which is also cloned as the static input injection $x_{\text{inj}}$. This $x_{\text{inj}}$ provides a consistent anchor during the subsequent iterative refinement. Next, the implicit equilibrium layer $f_{\text{imp}}$ refines the hidden state $z$ for a specified number of iterations $K_{\text{iter}}$, determined by the chosen per-scale schedule. In each iteration, $z$ is fused with $x_{\text{inj}}$ via a fusion projection Proj before passing through the transformer block. Unlike training, no gradients are computed, and the iteration count is strictly controlled to balance quality and latency. Finally, the converged state $z$ passes through the explicit post-layers $f_{\text{post}}$ to produce the next-scale token map $r_k$. This process repeats for all scales $k = 1 \ldots K$ to generate the full multi-scale image representation.

## B. Fusion Projection Architecture

To effectively integrate the static input injection $x_{\text{inj}}$ with the evolving hidden state $z$ within the implicit equilibrium layer, we employ a dedicated fusion projection module. As shown in Algorithm 1 and 2, this module, denoted as $\text{Proj}([z, x_{\text{inj}}])$, is applied at the beginning of each fixed-point iteration.

The fusion module is implemented as a two-layer Multi Layer Perceptron (MLP) that maps the concatenated input of dimension $2 \times D$ back to the model's embedding dimension $D$. Specifically, it consists of: 1. A linear expansion layer mapping from $2D$ to $2D$. 2. A GELU activation function (with tanh approximation). 3. A linear compression layer mapping from $2D$ to $D$.

Formally, given the current state $z$ and the input injection $x_{\text{inj}}$, the fused state $\tilde{z}$ is computed as:

$$\tilde{z} = W_2\left(\sigma\left(W_1[z, x_{\text{inj}}] + b_1\right)\right) + b_2 \tag{14}$$

where $[\cdot, \cdot]$ denotes concatenation along the channel dimension, $\sigma$ is the GELU activation, and $W_1 \in \mathbb{R}^{2D \times 2D}, W_2 \in \mathbb{R}^{D \times 2D}$ are learnable weights. This design stabilizes the fixed-point iteration by learning a non-linear combination of the persistent input context and the transient hidden dynamics, facilitating smoother convergence and better conditioning of the implicit operator.

## C. KV Cache Management

For VIAR, KV cache management within the implicit equilibrium layer requires a specialized approach to accommodate iterative refinement. Unlike standard autoregressive transformers that maintain a single growing cache per layer, the implicit layer must distinguish between different steps of the fixed-point iteration to ensure consistent attention contexts.

When KV caching is enabled, we structure the key-value storage as a dictionary indexed by the iteration count. During inference, for each fixed-point iteration step $t$, the model checks if a cache entry for $t$ exists. If it is the first time step $t$ is encountered for the current scale, a new cache entry is initialized. For subsequent tokens generated within the same iteration index $t$, the new keys and values are concatenated to this iteration-specific entry. This mechanism ensures that the self-attention operation at iteration $t$ attends only to the history corresponding to the same refinement stage $t$, effectively

*Table 4.* Quantitative comparison with other generative models on ImageNet $256 \times 256$. **VIAR** achieves superior performance with significantly fewer parameters compared to those generative models, Models with the suffix "-RE" used rejection sampling.

| MODELS | FID ↓ | IS ↑ | #PARAMS | #STEP |
|---|---|---|---|---|
| L-DiT-7B (ALPHA-VLLM, 2024) | 2.28 | 316.2 | 7.0B | 250 |
| VQGAN (ESSER ET AL., 2021) | 15.78 | 74.3 | 1.4B | 256 |
| VQGAN-RE (ESSER ET AL., 2021) | 5.20 | 280.3 | 1.4B | 256 |
| VITVQ (YU ET AL., 2021) | 4.17 | 175.1 | 1.7B | 1024 |
| VITVQ-RE (YU ET AL., 2021) | 3.04 | 227.4 | 1.7B | 1024 |
| RQ-TRAN. (LEE ET AL., 2022) | 7.55 | 134.0 | 3.8B | 68 |
| RQ-TRAN.-RE (LEE ET AL., 2022) | 3.80 | 323.7 | 3.8B | 68 |
| **VIAR (CFG=2.0)** | 2.35 | **330.7** | 770.9M | 10 |
| **VIAR (CFG=1.5)** | **2.16** | 300.1 | 770.9M | 10 |

isolating the dynamics of different fixed-point steps while preserving the autoregressive efficiency within each step. Standard explicit layers (pre- and post-layers) continue to use conventional single-stream concatenation.

## D. Ablation of Implicit and Explicit Architecture

We compare the deeply explicit stacked VAR against our VIAR. The tokenizer, data, and training protocol are identical. The parameter–FID frontier in Figure 8 shows that VIAR reaches competitive quality with markedly fewer parameters. In particular, VIAR achieves about 2.16 FID with 770.9M parameters, closely tracking a 2.0B-parameter explicit model at 2.08 FID. At intermediate scales, VIAR maintains a clear quality advantage over an equivalently sized explicit baseline, indicating better quality-per-parameter. When parameter or memory budgets are tight, VIAR provides a more favorable point on the quality–efficiency curve than explicit stacks. Explicit depth offers diminishing returns beyond 1–2B parameters, whereas VIAR attains comparable FID with substantially fewer parameters, making it a preferable choice for deployable models under strict size constraints.

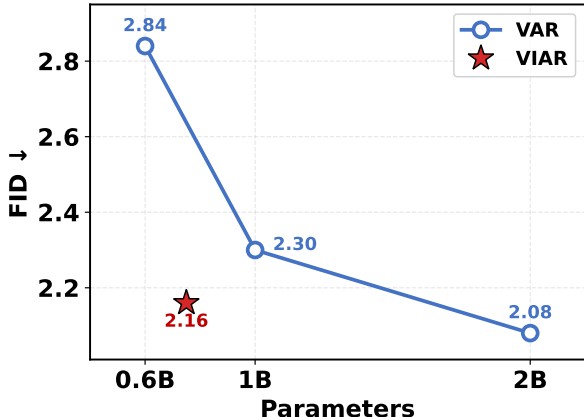

*Figure 8.* FID versus parameters for explicit VAR and our VIAR. VIAR attains near SOTA quality with substantially fewer parameters.

## E. Quantitative Comparison with Other Generative Models

Table 4 compares VIAR with a range of state-of-the-art generative models on ImageNet 256×256. VIAR (CFG=2.0) achieves an Inception Score of 330.7, significantly outperforming autoregressive baselines such as RQ-Transformer-RE and VQGAN-RE, as well as the diffusion-based L-DiT-7B. In terms of FID, VIAR (CFG=1.5) attains 2.16, which is superior to L-DiT-7B and all listed autoregressive models, including those using rejection sampling (e.g., ViTVQ-RE at 3.04). Crucially, VIAR achieves these results with only 770.9M parameters, which is approximately 11% of the size of L-DiT-7B and roughly half that of VQGAN, demonstrating exceptional parameter efficiency while maintaining top-tier generation quality.

## F. Comparison with Other Efficient VAR Methods

Recent works have explored different directions for improving the efficiency of visual autoregressive models. Among them, FastVAR (Guo et al., 2025) and ScaleKV (Li et al., 2025b) are closely related to our goal of reducing redundant computation in VAR-style generation. However, their mechanisms are largely orthogonal to VIAR. FastVAR and ScaleKV improve the efficiency of VAR-style generation from different perspectives. FastVAR reduces redundant computation at the token level by pruning less important tokens during inference, while ScaleKV reduces memory overhead by compressing the KV cache.

*Table 5.* Inference FLOPs and Training Wall-Clock Time.

| MODELS | TFLOPs | STEP TIME (MS) | TOTAL TIME (H) | #PARAMS |
|--------|--------|----------------|----------------|---------|
| VAR_D30 | 1.88 | 524.28 | 766.5 | 2010.0M |
| VIAR | 0.84 - 1.46 | 413.85 | 874.5 | 770.9M |

VIAR is orthogonal to these methods. Instead of applying a training-free acceleration strategy on top of an existing explicit backbone, VIAR redesigns the VAR backbone itself by replacing the deep explicit middle stack with an implicit equilibrium layer. This brings parameter reduction, constant-memory training, and flexible per-scale compute allocation. Therefore, VIAR can be viewed as a backbone-level efficiency improvement, while FastVAR and ScaleKV mainly focus on inference-time acceleration. These methods are complementary and could potentially be combined to further improve the efficiency of visual autoregressive generation.

## G. Inference FLOPs and Training Wall-Clock Time

For inference FLOPs, VIAR consistently requires less computation than the original VAR model under different iteration schedules. As shown in the Table 5, VAR requires 1.88 TFLOPs, while VIAR ranges from 0.84 TFLOPs to 1.46 TFLOPs depending on the selected schedule. Specifically, the lower bound 0.84 TFLOPs corresponds to the more aggressive $\text{VIAR}_{(10,1)}$ schedule, and the upper bound 1.46 TFLOPs corresponds to the more conservative $\text{VIAR}_{(10,10)}$ schedule. This shows that VIAR not only reduces parameter count, but also lowers inference computation. Moreover, by changing the inference schedule, VIAR can flexibly trade generation quality for computational cost within the same trained model.

VIAR is trained with S-JFB. During training, the implicit layer only performs $N$ no-gradient forward steps followed by $M$ gradient-enabled forward steps, with no additional operations beyond that. Here, $N$ is randomly sampled from 0–10 and $M$ from 1–12, so the average per-step training time of VIAR is actually lower than that of VAR. In Table 5, we report the per-step training time of VIAR and VAR on $8 \times \text{H100}$, as well as the total training time (Note that the total time for VAR is only an estimate: we did not fully train VAR to convergence ourselves, and the total training epochs are taken from the released VAR training script, so this should be regarded as a lower bound). More importantly, VIAR only needs to be trained once, and can then flexibly adjust the speed–quality tradeoff at inference time according to the compute budget. In contrast, supporting different inference-time compute budgets with VAR requires training separate models from scratch at different parameter scales.

## H. More Visual Results

We present more qualitative results produced by VIAR on ImageNet $256 \times 256$ in Figure 9. Under the default inference setting ($Con._{(10,10)}$, CFG=1.5), the model consistently generates semantically coherent images spanning a wide range of classes, including dynamic scenes (e.g., launch plumes and erupting volcanoes) and fine-grained objects (e.g., bird plumage, feathers, and rocky snow textures). Across categories, VIAR maintains long-range structural consistency (object shape, pose, and layout) and recovers high-frequency details (edges, textures, specular highlights) with low incidence of artifacts such as tiling, halos, or over-smoothing. These results complement the main-paper comparisons, illustrating that VIAR achieves strong visual quality and diversity while operating with reduced compute through its per-scale iteration control.

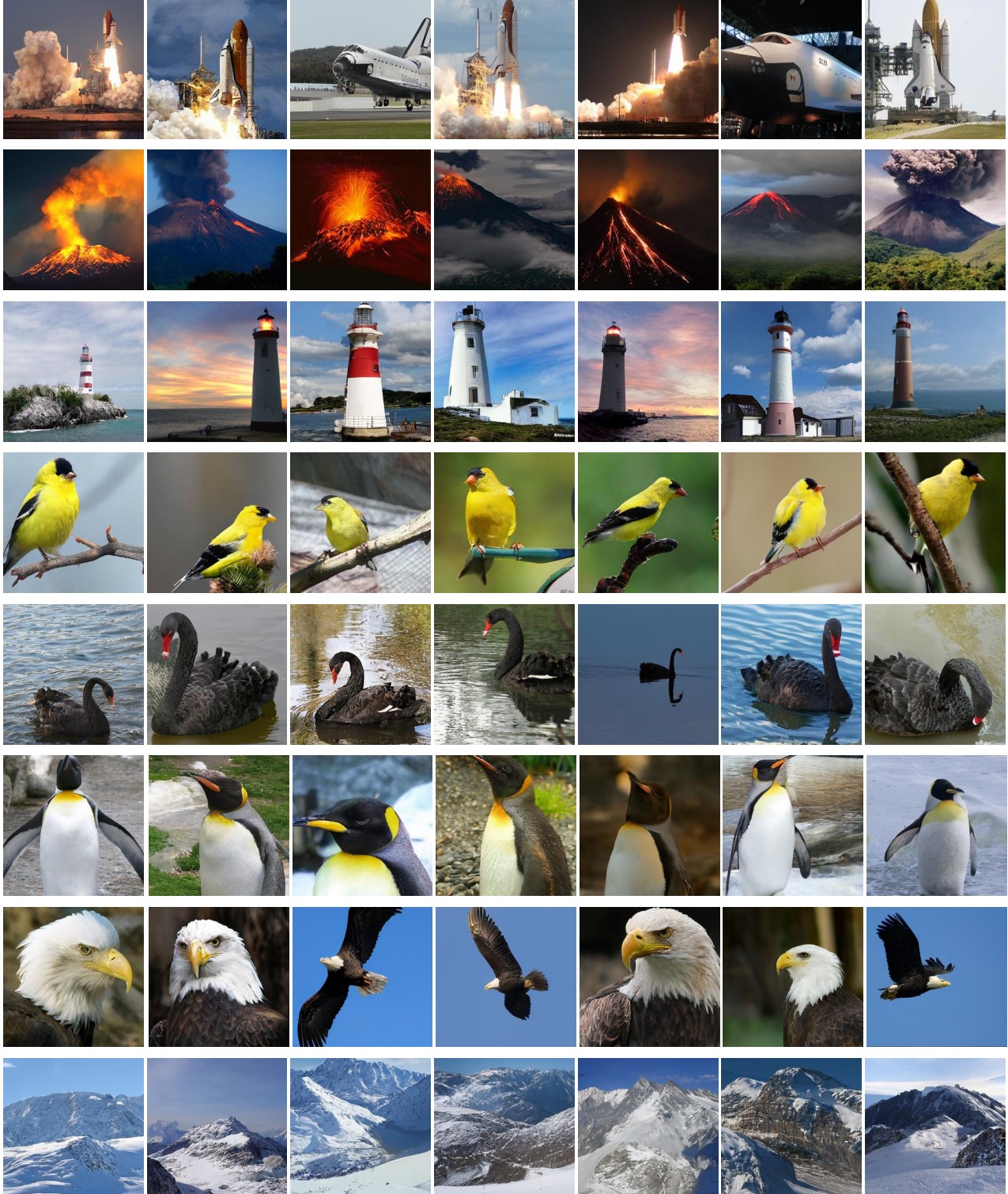

*Figure 9.* More visual results of VIAR on ImageNet. Samples are generated with a constant iteration count of 10 across all scales.

