# OpenReview forum: "Visual Implicit Autoregressive Modeling"
_ICML.cc/2026/Conference — ICML 2026 regular_

### Official Review · Reviewer_2JaH · 2026-03-08

**Soundness:** 3
**Presentation:** 3
**Significance:** 2
**Originality:** 2
**Overall Recommendation:** 3
**Confidence:** 4

**Summary:**

The paper proposes the method VIAR, which basically:
(1) Use the next scale prediction method for image generation, similar to VAR;
(2) For the architecture, apply Equilibrium based architecture rather than vanilla transformers.

The authors found that the choice of deep equilibrium transformer effectively saves parameter count and compute, with comparable generation performance, leading to faster inference speed on 4090 devices.

**Compliance With Llm Reviewing Policy:**

Affirmed.

**Final Justification:**

Strengths (Soundness & Clarity): The paper is generally well-written, and the empirical evaluations are sound. The experiments appear to be well-conducted, successfully demonstrating that applying Deep Equilibrium Models (DEQ) to Visual Autoregressive modeling (VAR) works as intended to reduce overall parameter counts while maintaining a similar FID/FLOPs tradeoff.

Weaknesses (Originality & Significance): My primary reservation lies in the novelty and overall impact of the contribution. Conceptually, applying DEQ to save parameters is a well-established ; thus, successfully applying it to VAR is somewhat expected and not surprising.

The authors' rebuttal unfortunately did not adequately address my main concern regarding the performance upper bound of this approach. Specifically, it remains unclear whether the proposed method can actually push or improve the Pareto frontier of the FID/FLOPs tradeoff, rather than merely matching the existing baseline with fewer parameters.

While the method is technically sound and the experiments validate its basic efficacy, the lack of significant innovation and the unresolved concern regarding the FID/FLOPs frontier limit the paper's overall significance. Because the rebuttal did not alleviate my primary concerns, it has reinforced my prior assessment. Therefore, I will maintain my current score.

**Key Questions For Authors:**

1. In table 3: what's the FLOPS for different schedules, and how do they compare to VAR? Are VIAR saving FLOPS, or only parameters?
2. As a result: if only parameter is saved, will the benefit go away or diminish at larger batch size, where the model parameters is not the bottleneck of memory?
3. Does VIAR use the same schedule during training and inference? Or the method allows a more flexible schedule at inference time, given the same training process?
4. If the same schedule is used: can the same parameter-saving effect be achieved by sharing the transformer weights for the middle layers?
5. The VAR d-30 results differ from the reported number in the VAR paper. In VAR paper, the FID of d-30 was 1.92. Why is there a difference?
6. Does VIAR achieve a better speed vs FID tradeoff, compared to the VAR family? For example. the FID seems comparable to VAR d-24 and VAR d-20. How does the speed compare to those?

**Limitations:**

yes

**Strengths And Weaknesses:**

Strengths:
1. Applying DeQ to VAR hasn't been explored before;
2. The inference speedup on 4090 devices is notable. The proposed way of saving parameters might be applicable in 4090 settings, where parameters can be a memory bottleneck.
3. Overall the paper is well written.

Weaknesses:

On the comparison with VAR:
1. The inference speed up mostly comes from saving the total number of parameters; it might be much less significant with larger batch sizes, when loading model parameter isn't the bottleneck, or when model parameters are quantized;
2. The generation FID of VIAR seems worse compared to original VAR d-30. As a result, the generation is slightly worse at slightly faster speed. It's unclear whether VIAR provides a better speed vs quality tradeoff. It might worth comparing speed with VAR d-20 and VAR d-24, which have comparable FID to VIAR.
3. There's no comparison of FLOPs. Not sure if the model saves FLOPs, or only parameter count.

On novelty:
The novelty is somewhat limited; since both DeQ for generation and VAR are already presented and well explored. It's not surprising that DEQ architectures can be applied to the VAR task.

Some minor weaknesses:
1. The paper claimed vanilla AR has n^6 complexity, which is not the case with KV-cache;
2. The VAR d-30 results differ from the reported number in the VAR paper. Why is this the case?
3. There're only 2 values of CFG. Sweeping the optimal CFG independently for every architecture will be a fairer comparison.

---

> ### Author Rebuttal · Authors · 2026-03-29
>
> Thank you for your detailed review and suggestion. We address your comments and questions in the following.
> 1. Comparison of FLOPs between VAR and VIAR.
> > Below we add a FLOPs comparison between VIAR under different schedules and VAR. The results show that VIAR saves not only parameters, but also FLOPs. We will add this FLOPs comparison to the main paper in the revised version to make this point clearer.
> |Model|GFLOPs|FID|
> |-|-|-|
> |VAR|1881.60|2.08|
> |VIAR_(10,10)|1458.52|2.16|
> |VIAR_(10,5)|1091.86|2.22|
> |VIAR_(5,5)|1046.44|2.27|
> |VIAR_(10,1)|839.64|2.43|
> 2. Latency comparison of VAR and VIAR under large batch size.
> > We provide the inference speedup results of VIAR under large batch sizes.
> |bs|64|128|192|
> |-|-|-|-|
> |VAR|1185.02 ms|2282.27 ms|OOM|
> |VIAR_(10,10)|961.06 ms|1857.01 ms|2753.67 ms|
> |VIAR_(10,5)|789.97 ms|1521.91 ms|2257.41 ms|
> |VIAR_(5,5)|760.59 ms|1471.28 ms|2179.88 ms|
> 3. The schedules during training and inference.
> > VIAR does not require the same schedule during training and inference. Specifically, during training, the number of implicit-layer iterations is randomly sampled so that the model can adapt to different iteration budgets. During inference, the same trained model can use more flexible per-scale schedules to allocate computation.
> 4. Differences from weight sharing.
> > We agree that weight sharing is a natural parameter-reduction baseline. However, VIAR’s gains are not just from reusing one set of weights, but from the implicit equilibrium formulation itself. In App. Sec. E, with matched tokenizer/data/training, VIAR (770.9M, FID 2.16) outperforms 1B VAR (2.30) and 0.6B VAR (2.84), while approaching 2B VAR (2.08). Unlike shared-weight explicit layers, VIAR also provides constant-memory training, fixed-point refinement, and flexible inference-time schedules within the same trained model.
> 5. The VAR d-30 results differ from the reported number in the VAR paper.
> > This discrepancy mainly comes from runtime environment and dependency versions differences. We use the publicly released VAR_d30 checkpoint from the VAR GitHub repository, rather than a model re-trained in the authors’ internal setup. Notably, even the released checkpoint differs slightly from the paper’s reported number: the paper reports FID 1.92, while the GitHub result is 1.97. More importantly, all results in our paper are evaluated under the same environment, pipeline, and hyperparameters, so the relative comparison is fair and consistent.
> 6. Comparison with the VAR family, including VAR_d24 and VAR_d20.
> > Appendix Sec. E and Fig. 8 already show the FID–parameter frontier, where VIAR is overall better than the VAR family in quality vs. model size. We further add the speed–FID tradeoff against VAR_d24 and VAR_d20: VIAR achieves better FID than VAR_d24 with fewer parameters and lower latency, and offers a clearly better quality regime than VAR_d20. More importantly, VIAR is trained once and can flexibly adjust the speed–quality tradeoff at inference via schedules, whereas each VAR model at a different parameter scale must be trained separately from scratch.
> |Model|params|lat(bs192)|FID|
> |-|-|-|-|
> |VAR_d20|0.6B|1770.75 ms|2.84|
> |VAR_d24|1B|2297.59 ms|2.30|
> |VIAR_(10,5)|770.9M|2257.41 ms|2.22|
> |VIAR_(5,5)|770.9M|2179.88 ms|2.27|
> |VIAR_(10,1)|770.9M|1919.37 ms|2.43|
> 7. Clarify the novelty of VIAR.
> > Thank you for the comment. Our point is that VIAR is not simply “DEQ + VAR.” To our knowledge, this is the first work to integrate an equilibrium formulation into VAR’s next-scale prediction paradigm. This is nontrivial because VAR involves coarse-to-fine conditioning across scales, within-scale parallel prediction, and interactions with KV cache, and fixed-point refinement. Rather than a plug-in replacement, VIAR redesigns the middle computation in each next-scale step with an implicit layer while preserving VAR’s multi-scale parallel generation structure. VIAR also turns VAR’s fixed explicit depth into a per-scale iteration knob, enabling adaptive compute, reduced KV cache, and flexible quality–efficiency tradeoffs.
> 8. The paper claimed vanilla AR has $O(n^6)$ complexity, which is not the case with KV-cache.
> > Thank you for the correction. You are right that stating vanilla AR as $O(n^6)$ is imprecise. That result assumes no KV cache, i.e., recomputing full self-attention over the entire prefix at every step. But the $O(n^4)$ complexity for VAR in the original paper is also derived under the same full-recomputation assumption, not KV-cached decoding. In the revised version, we will clearly distinguish between naive AR without KV cache and standard AR decoding with KV cache, to avoid confusion and make the comparison more precise.
> 9. Sweeping CFG separately for each model will be a fairer comparison.
> > We reported only two CFG values to stay consistent with the original VAR paper. We have further checked the best CFG for each architecture: VAR achieves its best FID 2.05 at CFG 2.0, while VIAR achieves its best FID 2.15 at CFG 1.4.

---

> > ### Author Rebuttal · Reviewer_2JaH · 2026-04-04
> >
> > I thank the authors for the rebuttal. The rebuttal is clear and provides useful signals. However, I would like to maintain my score, since:
> >
> > 1. Based on the response to question 6, the improvement of FLOPS / FID tradeoff seems pretty marginal.
> > 2. In response to question 7, I don't see how the method differ from "DEQ + VAR". All the descriptions here:
> >
> > > Rather than a plug-in replacement, VIAR redesigns the middle computation in each next-scale step with an implicit layer while preserving VAR’s multi-scale parallel generation structure. VIAR also turns VAR’s fixed explicit depth into a per-scale iteration knob, enabling adaptive compute, reduced KV cache, and flexible quality–efficiency tradeoffs.
> >
> > are applicable to DEQ + VAR.
> >
> > As a result, I still feel this method is basically "DEQ + VAR", where the flexibility gives slightly better frontier of FLOPS vs FID. I will maintain my score.

---

> > > ### Author Response · Authors · 2026-04-04
> > >
> > > We thank the reviewer for the follow-up feedback. Below, we reiterate the efficiency and innovation of VIAR.
> > > 1. The improvement in FLOPs/FID tradeoff is one aspect, but VIAR also significantly reduces the parameter count: it reduces 61.6% compared to VAR_d30 and 24.7% compared to VAR_d24, while achieving a FID that is close to VAR_d30 and better than VAR_d24. Additionally, VIAR significantly outperforms VAR_d20 in FID. More importantly, **VIAR only needs to be trained once, and can flexibly adjust schedules at inference time to achieve these results**. In contrast, VAR requires training separate models from scratch for different parameter scales.
> > > 2. VIAR integrates the implicit layer iteration concept from DEQ into the VAR generation process, but the specific implementation requires adapting to various features of VAR. This is nontrivial because VAR involves coarse-to-fine conditioning across scales, within-scale parallel prediction, and interactions with KV cache, and fixed-point refinement. Additionally, VIAR is the first work to integrate the implicit layer iteration concept into VAR.

---

### Official Review · Reviewer_VuBT · 2026-03-08

**Soundness:** 3
**Presentation:** 3
**Significance:** 4
**Originality:** 4
**Overall Recommendation:** 4
**Confidence:** 3

**Summary:**

This paper addresses the computational and memory bottlenecks of the Visual Autoregressive Modeling (VAR) paradigm (i.e., next-scale prediction). While standard VAR effectively resolves the structural mismatch of raster-scan next-token AR models by predicting token maps from coarse to fine, its explicitly stacked deep transformer layers lead to massive parameter counts, large KV cache footprints, and fixed inference costs, particularly at high resolutions. To overcome these limitations, the authors propose Visual Implicit Autoregressive Modeling (VIAR). VIAR replaces the deep explicit middle layers of the VAR architecture with a single implicit equilibrium layer (inspired by Deep Equilibrium Models, DEQs), sandwiched between shallow explicit pre- and post-layers. During training, VIAR employs Stochastic Jacobian-Free Backpropagation (S-JFB) to ensure a strictly constant memory footprint regardless of the effective depth. During inference, VIAR exposes an adaptive "iteration knob", allowing the model to dynamically allocate more iterations to coarse scales (for global structure) and fewer to fine scales (where refinement saturates quickly). Extensive experiments on ImageNet 256×256 demonstrate that VIAR reduces the parameter count by 61.6% and peak inference memory by up to 42.0%, while maintaining or even slightly improving the generation fidelity (FID 2.16, sFID 8.07) and zero-shot editing capabilities compared to the standard VAR baseline.

**Compliance With Llm Reviewing Policy:**

Affirmed.

**Key Questions For Authors:**

1. Training Wall-Clock Time: Could you provide a direct comparison of the overall training time (e.g., GPU hours/days to reach convergence) between VIAR and the VAR-d30 baseline? How much computational overhead does the implicit solver and S-JFB introduce per training step?
2. KV Cache Mechanics: The paper mentions that VIAR "significantly reduces the generated KV cache." Could you clarify how the KV cache is managed within the implicit layer across the $N$ iterations? Does the model only cache the final equilibrium state $z_k^$ for the cross-attention of subsequent scales?
3. Scalability: Have you conducted any preliminary experiments applying VIAR to higher resolutions (e.g., 512x512) or text-to-image generation? Since VAR scales cleanly, does the implicit equilibrium formulation maintain stability at larger scales or more complex conditionings?

**Limitations:**

No. The authors do not adequately discuss technical limitations. The authors should explicitly discuss the training wall-clock time overhead introduced by the implicit solver (S-JFB), as this is a known bottleneck for DEQ-based models despite the memory savings.

**Strengths And Weaknesses:**

Strengths:
1. The methodology is technically rigorous and well-motivated. The use of S-JFB for training implicit models is a mature technique, and the convergence analysis in Figure 3 compellingly justifies the design choice of reducing iteration counts at larger scales. The empirical evaluations are thorough in terms of comparing generation quality (FID, sFID, IS) against parameter count, memory footprint, and throughput.
2. The paper is exceptionally well-written and logically organized. Figures 1 and 2 brilliantly encapsulate the core architectural differences and resource savings between VAR and VIAR. The narrative naturally flows from the limitations of fixed-depth VAR to the theoretical background of DEQ, culminating in the VIAR design.
3. The memory and latency bottlenecks of high-resolution AR generation are pressing issues for real-world deployment (e.g., edge devices or consumer GPUs). By successfully decoupling the model's depth from its memory footprint and offering a zero-shot, tunable quality-efficiency knob during inference, VIAR offers immense practical utility.
Weaknesses:
1. Training Wall-Clock Time: While the authors emphasize the constant training memory of VIAR (Figure 6), they completely omit the training wall-clock time. Solving for the equilibrium state and unrolling S-JFB typically introduces substantial computational overhead per training step compared to a standard forward pass. It is crucial to clarify whether VIAR takes significantly longer to train to convergence than the baseline VAR.
2. Scale of Experiments: The evaluation is limited to class-conditional ImageNet at 256x256. Modern AR models (e.g., LlamaGen, Lumina-mGPT, VAR-d30) are increasingly evaluated on higher resolutions (512x512) or open-domain text-to-image datasets. Demonstrating VIAR's scalability to larger resolutions or cross-modal conditions would significantly strengthen the paper's claims.

---

> ### Author Rebuttal · Authors · 2026-03-29
>
> Thank you for your detailed review and suggestion. We address your comments and questions in the following.
> 1. Training Wall-Clock Time: Could you provide a direct comparison of the overall training time (e.g., GPU hours/days to reach convergence) between VIAR and the VAR-d30 baseline? How much computational overhead does the implicit solver and S-JFB introduce per training step?
> > VIAR is trained with S-JFB. During training, the implicit layer only performs N no-gradient forward steps followed by M gradient-enabled forward steps, with no additional operations beyond that. Here, N is randomly sampled from 0–10 and M from 1–12, so the average per-step training time of VIAR is actually lower than that of VAR. Below, we report the per-step training time of VIAR and VAR on 8×H100, as well as the total time to convergence (Note that the total time for VAR is only an estimate: we did not fully train VAR to convergence ourselves, and the total training epochs are taken from the released VAR training script, so this should be regarded as a lower bound). More importantly, VIAR only needs to be trained once, and can then flexibly adjust the speed–quality tradeoff at inference time according to the compute budget. In contrast, supporting different inference-time compute budgets with VAR requires training separate models from scratch at different parameter scales.
> |Model|step(ms)|hours|
> |-|-|-|
> |VAR_d30|524.28|766.5(lower-bound)|
> |VIAR|413.85|874.5|
> 2. KV Cache Mechanics: The paper mentions that VIAR "significantly reduces the generated KV cache." Could you clarify how the KV cache is managed within the implicit layer across the iterations? Does the model only cache the final equilibrium state $z_k^*$ for the cross-attention of subsequent scales?
> > Thank you for the question. We already describe this in Appendix Section C (KV Cache Management) and will clarify it further in the main text.
> Specifically, VIAR does not cache only the final equilibrium state $z_k^*$ inside the implicit layer. Because the implicit layer involves fixed-point iterations, different iteration steps correspond to different refinement stages. Therefore, we organize the KV cache as a dictionary indexed by iteration step t. During inference, the t-th iteration only reads and updates the cache associated with that same t, ensuring that self-attention at iteration t only attends to history from the same refinement stage, without mixing states across different iterations. In contrast, the pre-layers and post-layers still use the standard single-stream KV cache.
> This also explains what we mean by saying that VIAR “significantly reduces the generated KV cache.” In practice, the implicit layer in VIAR usually converges within 8–10 iterations at inference time, which effectively compresses the KV cache corresponding to the original 20 explicit middle layers into at most about 10 iteration-specific cache entries. In other words, VIAR is not only better aligned with fixed-point refinement in how the cache is organized, but also yields a substantially smaller overall cache than the original explicit VAR.
> 3. Scalability: Have you conducted any preliminary experiments applying VIAR to higher resolutions (e.g., 512x512) or text-to-image generation? Since VAR scales cleanly, does the implicit equilibrium formulation maintain stability at larger scales or more complex conditionings?
> > Thank you for the question. We agree that extending VIAR to higher-resolution generation (e.g., 512×512) and more complex conditional settings such as text-to-image generation is an important and meaningful direction, and it is one of our planned future directions. We will systematically study the scalability and stability of VIAR in these larger-scale and more complex generation settings in future work.
> At present, we have not fully included such extensions in this paper for two practical reasons. First, training and evaluation at larger scales are significantly more expensive, especially for high-resolution generation. Second, some large-scale text-to-image VAR systems (e.g., Infinity) rely on training data that are not publicly available, making a fully comparable extension difficult to complete quickly.
> That said, we have already conducted some preliminary exploration. Our current results show that under larger-scale generation settings, the training loss of VIAR still decreases stably, without obvious optimization instability. This provides initial evidence that the implicit equilibrium formulation remains stable at larger scales and has the potential to extend to higher-resolution and more complex conditional generation tasks.

---

> > ### Author Rebuttal · Reviewer_VuBT · 2026-04-03
> >
> > The training time data adequately addresses my main concern — VIAR's per-step cost is actually lower than VAR, and the argument about training once vs. training separate models is convincing. The KV cache explanation is also clear. On scalability, I consider the lack of higher-resolution or T2I results a remaining limitation, but this does not change my overall assessment. I maintain my score of 4 (Weak Accept).

---

> > > ### Author Response · Authors · 2026-04-04
> > >
> > > We sincerely thank the reviewer for the positive feedback. We are pleased that our response has addressed all concerns, and we will incorporate these additions into the final version.

---

### Official Review · Reviewer_GGCx · 2026-03-09

**Soundness:** 3
**Presentation:** 3
**Significance:** 2
**Originality:** 2
**Overall Recommendation:** 4
**Confidence:** 4

**Summary:**

The paper proposes Visual Implicit Autoregressive Modeling (VIAR) by integrating deep equilibrium architecture into Visual Autoregressive Modeling (VAR). Using deep equilibrium layer has the benefit of reduced training memory and dynamic adjustment of compute at inference. On ImageNet 256px, VIAR achieves comparable performance to VAR while using only 38.4% of the parameters of VAR.

**Compliance With Llm Reviewing Policy:**

Affirmed.

**Final Justification:**

I am leaning toward the acceptance of the paper. The research demonstrates the DEQ can be integrated in VAR and the reported performance is valuable for the community for guiding future research.

**Key Questions For Authors:**

1. The FID/IS evaluation is conducted on 50k ImageNet validation set. But most of the diffusion papers (SiT, DiT) have an established convention of reporting FIDs on the training set. Did VAR also support FID on validation set?

2. Is there work exploring DEQ in diffusion models? If so, it should be included in related work. If not, have you thought about exploring DEQ in diffusion models? That seems like a more significant area of research?

3. I would recommand putting FIDs as the first column in all tables, as FIDs are the main metrics today's research count on instead of inception score. The same goes to the text. The paper tends to refer to the inception score as the primary metric.

**Limitations:**

VIAR achieves slightly worse FID (2.16) compared to VAR (2.05) while using 38.4% of the parameter. I would be nice to show at what percentage of parameters does VIAR surpass/reach the FID of VAR. This is only a small issue.

**Strengths And Weaknesses:**

1. On originality, the paper explores integrating deep equilibrium model layers (DEQ), which is proposed by prior work [1], with visual autoregressive modeling (VAR), a next-scale autoregressive generation method also proposed by prior work [2]. The paper does not propose innovation in theory or formulation. Also I am not convinced that the technique or benefits of integrating DEQ is unique to VAR. For example, DEQ can also be integrated into diffusion models in the same way, so that different timesteps or different resolutions may dynamically trade off the compute at inference. I do believe the work is valuable as it shows that DEQ and be integrated in VAR, but it does not show strong originality in the work.

2. The proposed method is sound. In the related work section the authors have pointed out prior work indicating the redundancy in model parameters used in VAR, which motivates the exploration of DEQ.

3. The presentation of the work is clear.

4. Significance is fair. Both VAR and VIAR has yet to surpass diffusion/flow-matching models in FIDs. The proclaimed benefit of efficiency is also weak when compared to distilled/consistency models. It is good to see continued improvements in autoregressive modeling but it is not as popularized as flow-based approaches.

[1] Deep equilibrium models
[2] Visual Autoregressive Modeling: Scalable Image Generation via Next-Scale Prediction

---

> ### Author Rebuttal · Authors · 2026-03-29
>
> Thank you for your detailed review and suggestion. We address your comments and questions in the following.
> 1. About FID/IS evaluation.
> > For FID/IS, we evaluate on the 50k ImageNet validation set. This is also consistent with DiT, which reports FID-50K on the validation set rather than the training set. We use the validation set because it better reflects generalization to unseen data, which is the more standard and practically meaningful evaluation protocol for generative models. VAR follows the same convention and also reports FID on the validation set. While some works may use different settings, we believe validation-set evaluation is the more common and appropriate choice for assessing real model performance.
> 2. Is there work exploring DEQ in diffusion models? If so, it should be included in related work. If not, have you thought about exploring DEQ in diffusion models? That seems like a more significant area of research?
> > Several prior works have already explored the application of DEQs within the context of Diffusion Models. In the related work section of our paper, we have included FPDM [1], the work most relevant to ours, to highlight some of the explorations and advancements involving DEQs in Diffusion Models. We acknowledge the potential of DEQs in Diffusion Models and believe that this direction warrants further investigation. We have also referenced these existing studies in our work to provide a comprehensive background.
> [1] Fixed Point Diffusion Models
> 3. I would recommand putting FIDs as the first column in all tables, as FIDs are the main metrics today's research count on instead of inception score. The same goes to the text. The paper tends to refer to the inception score as the primary metric.
> > Thank you for the valuable suggestion. We fully agree that FID is the more important evaluation metric in current research. In the revised version, we will reorder the tables and place FID as the first column in all tables. We will also revise the text accordingly to ensure that FID is discussed as the primary evaluation metric.
> 4. VIAR achieves slightly worse FID (2.16) compared to VAR (2.05) while using 38.4% of the parameter. I would be nice to show at what percentage of parameters does VIAR surpass/reach the FID of VAR.
> > Thank you for the valuable suggestion. We understand the importance of comparing VIAR and VAR in terms of FID under different parameter budgets, and we agree that this would be a very meaningful experiment for demonstrating VIAR’s parameter efficiency. For the current submission, we have already provided a preliminary comparison of parameter efficiency between VAR and VIAR in Appendix Section E and Figure 8.
> However, due to training-time constraints, we are not able to complete this additional experiment during the rebuttal period. We plan to include this analysis in future work and in a revised version, to more clearly show how VIAR approaches or surpasses VAR’s FID as the parameter budget varies.

---

> > ### Author Rebuttal · Reviewer_GGCx · 2026-03-31
> >
> > Incorrect. DiT, SiT, and all other major works follow the ADM evaluation setup, which is FID measured on the ImageNet training set, not the validation set. Specifically, the generator model samples 50k images with uniform random class labels. These 50k images are compared against the statistics calculated on the entire ImageNet training set.
> >
> > Cite from ADM paper:
> > "To ensure consistent comparisons, we use the entire **training** set as the reference batch, and evaluate metrics for all
> > models using the same codebase."
> >
> > Cite from SiT paper:
> > "We calculate FID scores between generated images (10K or 50K) and all available real images in ImageNet **training** dataset."
> >
> > Cite from DiT paper:
> > "all values reported in this paper are obtained by exporting samples and using ADM’s TensorFlow evaluation suite"
> >
> > After reviewing VAR, which is the baseline method that this work is comparing to, I believe the VAR paper only provides the validation set 50k as a "reference lower-bound", but the model is still evaluated against the **training set** using the ADM suite.
> >
> > Code: https://github.com/FoundationVision/VAR/blob/78b95394fc5896192e3a003e4b295f8ea743c48f/README.md?plain=1#L156
> >
> > I think this is quite a problem for this paper, as the FID is not computed correctly following past convention.
> > Can the authors confirm how the VAR FID presented in this paper is computed? Is it also against validation set?
> >
> > * [DiT]: Scalable Diffusion Models with Transformers
> > * [SiT]: SiT: Exploring Flow and Diffusion-based Generative Models with Scalable Interpolant Transformers
> > * [VAR]: Visual Autoregressive Modeling: Scalable Image Generation via Next-Scale Prediction
> > * [ADM]: Diffusion Models Beat GANs on Image Synthesis

---

> > > ### Author Response · Authors · 2026-04-01
> > >
> > > Thank you for the correction. We re-checked the FID evaluation details in DiT, SiT, and ADM, and indeed, FID was evaluated on the training set. We were previously misled by the description in Figure 3 of the VAR paper mentioning the “validation set 50k as a reference lower-bound.” After carefully reviewing VAR’s FID evaluation details, we confirmed that VAR also evaluates FID against the full training dataset:
> > > https://github.com/openai/guided-diffusion/blob/main/evaluations/README.md?plain=1#L9
> > > Since VIAR follows the same evaluation setting and protocol as VAR, VIAR’s FID is also evaluated on the training set. We will correct this mistake in the revised version and clarify the evaluation setting and protocol explicitly.

---

### Official Review · Reviewer_75av · 2026-03-11

**Soundness:** 3
**Presentation:** 3
**Significance:** 3
**Originality:** 3
**Overall Recommendation:** 4
**Confidence:** 4

**Summary:**

This paper proposes Visual Implicit Autoregressive Modeling (VIAR), replacing the deep explicit stacks in VAR models with a single implicit equilibrium layer. Trained via Stochastic Jacobian-Free Backpropagation, VIAR maintains a constant training memory footprint. During inference, a tunable per-scale iteration knob enables flexible, budget-aware compute allocation. On ImageNet 256×256, VIAR achieves competitive generation quality (FID 2.16) using only 38.4% of the parameters of a 2B VAR baseline. Furthermore, it significantly improves deployment efficiency—more than halving peak GPU memory and doubling throughput—while demonstrating strong zero-shot editing capabilities.

**Compliance With Llm Reviewing Policy:**

Affirmed.

**Final Justification:**

The rebuttal addressed most of my concerns and improved my overall assessment of the paper. However, the backbone coverage is still somewhat limited. I therefore maintain my final recommendation as Weak Accept, and raise my confidence to 4.

**Key Questions For Authors:**

1.	Could the authors compare VIAR against an explicit baseline under matched compute, for example an adaptive-skipping or early-exit VAR with a total effective depth comparable to 5 + $K_{iter} + 5$? This would help clarify whether the gain comes specifically from implicit fixed-point refinement rather than simply from using more iterative computation.
2.	How sensitive is VIAR to the S-JFB hyperparameters, especially the iteration counts N and M? A clearer ablation on training stability and final generation quality would help assess the robustness of the proposed training scheme, and it would also be useful to compare against a simpler 1-step JFB variant.
3.	How does VIAR compare with recent VAR-specific efficiency methods, such as FastVAR, on the quality-throughput trade-off? Since these methods also target redundant computation at high-resolution scales, such a comparison would better position the contribution of the implicit equilibrium design.

**Limitations:**

No. The paper does not explicitly discuss its limitations or potential negative societal impact.

**Strengths And Weaknesses:**

**Strengths**

1.	The paper is well motivated and tackles a practical limitation of next-scale autoregressive generation, namely the fixed-depth computation overhead in explicit VAR models. Introducing an implicit equilibrium formulation into this setting is a novel and meaningful idea.
2.	The method is clearly designed and well aligned with the motivation. The combination of an implicit core, Jacobian-free training, and per-scale iteration control provides a coherent framework for reducing memory cost while enabling flexible inference-time computation.
3.	The empirical results are strong and show clear gains in both efficiency and generation quality. The paper demonstrates favorable trade-offs in parameter count, memory usage, throughput, and image quality, supporting the practical value of the proposed approach.


**Weaknesses**

1.	The experimental comparison is still somewhat limited. The quantitative evaluation relies too heavily on the original VAR baseline. To better assess VIAR's relative advantages, the authors should include comparisons against stronger, more recent next-scale autoregressive methods such as FastVAR[1], ScaleKV [2] or different backbones like Infinity.
2.	Although VIAR improves throughput largely by reducing memory usage, the classic challenge of implicit models is the cost of fixed-point iteration. While Figure 3 suggests that 5 to 10 iterations are often sufficient, repeated iterations still introduce nontrivial computation compared with a single explicit forward pass. The paper reports throughput and memory, but does not analyze single-image latency, FLOPs, especially in more compute-bound regimes. Such analysis would better clarify the practical efficiency of the method.
3.	The paper emphasizes constant training memory, but does not report wall-clock training time or step-time comparisons against explicit VAR. Since implicit methods often trade memory savings for additional solver overhead, a clearer accounting of training cost would make the efficiency claims more complete.
4.	The ablation coverage is still limited relative to the complexity of the overall design. While the paper studies iteration schedules and convergence behavior, it does not fully disentangle how much of the gain comes from the implicit equilibrium formulation itself, the per-scale iteration schedule, or other architectural choices.


[1] Fastvar: Linear visual autoregressive modeling via cached token pruning.

[2] Memory-efficient visual autoregressive modeling with scale-aware kv cache compression.

---

> ### Author Rebuttal · Authors · 2026-03-29
>
> Thank you for your detailed review and suggestion. We address your comments and questions in the following.
> 1. Comparisons against efficient methods such as FastVAR, ScaleKV or different backbones like Infinity.
> >Thank you for the suggestion. Infinity is not directly comparable here, its training data/recipe are not fully public, making a fair backbone-level comparison difficult. We agree that extending VIAR to more complex text-to-image backbones is a meaningful direction, and it is one of the directions we plan to explore in future work.
> FastVAR and ScaleKV are closer, but they are orthogonal to VIAR: FastVAR reduces token-level redundancy, and ScaleKV compresses KV cache during inference, whereas VIAR redesigns the VAR backbone itself with an implicit equilibrium layer, yielding parameter reduction, constant-memory training, and flexible per-scale compute allocation. In short, VIAR is not just another inference-time acceleration trick, it changes the backbone computation itself, and can potentially be combined with FastVAR/ScaleKV.
> 2. Analyze the latency and FLOPs for a single image in compute-bound scenarios.
> >We have added latency and FLOPs analysis for a single image (bs=1) using a single RTX 3090:
> |Model|lat(ms)|GFLOPs|
> |-|-|-|
> |VAR|356.34|1881.60|
> |VIAR_(10,10)|276.01|1458.52|
> |VIAR_(10,5)|228.87|1091.86|
> |VIAR_(5,5)|206.84|1046.44|
> 3. Comparing the single-step training time and wall-clock training time of VIAR and VAR.
> >VIAR is trained with S-JFB: the implicit layer performs only N no-grad forwards followed by M grad-enabled forwards, with no extra operations, the average per-step cost is in fact lower than VAR. Below we report step time on 8×H100 and an estimated total time to convergence. The total VAR time is only a lower-bound estimate according to the released VAR training script, since we did not fully retrain VAR to convergence. Moreover, VIAR needs to be trained only once, and can then flexibly adjust the speed–quality tradeoff at inference via schedules, whereas VAR would require training separate models for different compute budgets.
> |Model|step(ms)|hours|
> |-|-|-|
> |VAR_d30|524.28|766.5(lb)|
> |VIAR|413.85|874.5|
> 4. The impact of implicit equilibrium layer design, per-scale iteration schedules, and architectural choices on results.
> >These components play different roles. The implicit formulation provides the main structural gains: replacing the deep explicit middle stack with one equilibrium layer improves parameter efficiency, introduces fixed-point refinement. In Appendix Sec. E (implicit vs. explicit ablation), with tokenizer, data, and training protocol matched, VIAR achieves a better parameter–FID frontier than explicit VAR.
> By contrast, the per-scale schedule mainly controls inference-time compute allocation within the same trained model, trading FID for latency/memory/FLOPs. Input injection is standard in DEQ/implicit models. More exhaustive ablations over other architectural and training details would require training many models from scratch, which is prohibitively expensive.
> Thus, our current experiments already separate the roles of the two key components. We acknowledge that more granular ablation studies regarding architecture would be valuable, and we intend to incorporate these into the subsequent version.
> 5. Compare VIAR against an explicit early-exit VAR baseline under matched compute.
> >We have added an experiment comparing VIAR against an explicit early-exit VAR baseline under matched compute:
> |Model|FID|
> |-|-|
> |VAR(early-exit)|83.98|
> |VIAR_(10,10)|2.16|
> 6. Ablation study on N and M during S-JFB, and compare with 1-step JFB.
> >Training multiple models from scratch is very costly. Thus, in this work we chose these training hyperparameters heuristically. We set N=10, M=12 so that VIAR’s maximum number of layer forwards stays comparable to VAR, keeping inference-time implicit iterations in a practical range (about 1–20), balancing training cost, flexibility, and stability.
> We add a 20-epoch comparison between S-JFB and 1-step JFB: 1-step JFB shows noticeably slower loss reduction, indicating weaker convergence. Meanwhile, setting a smaller value of M in S-JFB leads to slower loss decay and consequently slower convergence, the same trend also holds for N.
> |Method|l_mean|l_tail|
> |-|-|-|
> |1-step JFB|6.372|5.741|
> |S-JFB(M=3)|6.361|5.730|
> |S-JFB(M=6)|6.341|5.695|
> |S-JFB(M=12)|6.301|5.636|
> 7. Comparison with FastVAR.
> >FastVAR mainly gains speed from Cached Token Pruning at large scales (e.g., 1024×1024). At small scales, its extra operations (high-frequency token selection and interpolation) can offset the savings. In contrast, VIAR achieves acceleration by restructuring the backbone computation, thereby reducing the total layer forwards. This yields consistent speedups across both small and large scales. We also ported FastVAR to VAR_d30 and compare it against both original VAR and VIAR below.
> |Model|lat(ms)|
> |-|-|
> |VAR|356.34|
> |VAR+FastVAR|354.19|
> |VIAR_(10,10)|276.01|

---

> > ### Author Rebuttal · Reviewer_75av · 2026-04-01
> >
> > The rebuttal addressed most of my concerns. If the authors view methods such as FastVAR as not directly comparable to VIAR, but potentially compatible with it, could they provide an experiment applying such a training-free acceleration method on top of VIAR? I think FastVAR is designed for explicit next-scale VAR inference with cached token pruning and token restoration, whereas VIAR performs iterative per-scale refinement through an implicit equilibrium layer, so their compatibility is not yet obvious to me.

---

> > > ### Author Response · Authors · 2026-04-02
> > >
> > > FastVAR accelerates inference through token pruning, whereas VIAR accelerates by reducing the number of layer forward passes, so the two methods are naturally orthogonal. However, because FastVAR introduces extra operations such as high-frequency token selection and interpolation, its speedup mainly comes from pruning tokens at large scales (e.g., 1024×1024). Under the VAR 256×256 setting, FastVAR brings only a very small speedup, we already showed the effect of applying FastVAR to VAR_d30 in our response to Question 7. We further add the effect of applying FastVAR to VIAR below:
> > >
> > > |Model|latency(ms)|
> > > |-|-|
> > > |VAR|356.34|
> > > |VAR+FastVAR|354.19|
> > > |VIAR|276.01|
> > > |VIAR+FastVAR|274.64|
> > >
> > > In summary, FastVAR and VIAR are methodologically compatible, but because the extra overhead of FastVAR (e.g., high-frequency token selection and interpolation) is relatively large, the additional speedup from combining FastVAR with either VAR or VIAR is very limited in the current 256×256 setting.

---

### Decision · Program_Chairs · 2026-04-30

**Decision:**

Accept (regular)

**Comment:**

This submission received mixed ratings (3 Weak Accepts and 1 Weak Reject). The positive reviewers acknowledged that successfully adapting DEQ to the Visual Autoregressive (VAR) paradigm reduces parameter and memory usage, representing practical value. However, #2JaH raised a primary contention that the core novelty is limited, as the method is essentially an expected application of DEQ + VAR that produces only marginal improvements in the FLOPs/FID Pareto frontier.

The rebuttal was partially successful; #75av raised their score from negative to Weak Accept, but with reservations regarding the backbone coverage. #GGCx and #VuBT both maintained their positive scores, but also expressed reservations regarding the improvement over VAR and the lack of high-resolution results. #2JaH remained unconvinced regarding the novelty, still considering that applying DEQ to VAR lacks a fundamental contribution.

The paper falls exactly on the decision boundary. The AC initiated a post-rebuttal discussion, and all three positive reviewers (#75av, #VuBT, #GGCx) still expressed supportive views after reading #2JaH's comments, considering that the paper has strong practical utility.

The AC reviewed the paper, the rebuttal, and the reviewers' comments. Though the AC agrees with #2JaH that applying an existing technique to another indeed mitigates the theoretical contribution, given the broad impact of VAR models and the successful alleviation of the parameter bottleneck by the proposed method, the paper is worth presenting to the community.